# Age at Natural Menopause and Blood Pressure Traits: Mendelian Randomization Study

**DOI:** 10.3390/jcm10194299

**Published:** 2021-09-22

**Authors:** Zayne M. Roa-Díaz, Eralda Asllanaj, Hasnat A. Amin, Lyda Z. Rojas, Jana Nano, Mohammad Arfan Ikram, Fotios Drenos, Oscar H. Franco, Raha Pazoki, Pedro Marques-Vidal, Trudy Voortman, Taulant Muka

**Affiliations:** 1Institute of Social and Preventive Medicine (ISPM), University of Bern, 3012 Bern, Switzerland; zayne.roadiaz@ispm.unibe.ch (Z.M.R.-D.); oscar.franco@ispm.unibe.ch (O.H.F.); 2Graduate School for Health Sciences, University of Bern, 3012 Bern, Switzerland; 3Department of Epidemiology, Erasmus MC, University Medical Center, 3015 GD Rotterdam, The Netherlands; eralda.asllani@gmail.com (E.A.); m.a.ikram@erasmusmc.nl (M.A.I.); trudy.voortman@erasmusmc.nl (T.V.); 4Department of Life Sciences, College of Health, Medicine and Life Sciences, Brunel University London, Uxbridge UB8 3PH, UK; Hasnat.Amin@brunel.ac.uk (H.A.A.); Fotios.drenos@brunel.ac.uk (F.D.); raha.pazoki@brunel.ac.uk (R.P.); 5Nursing Knowledge Research and Development Group(GIDCEN), Fundación Cardiovascular de Colombia, Floridablanca 681004, Colombia; lydar7@hotmail.com; 6Institute of Epidemiology, Helmholtz Zentrum München, 85764 Neuherberg, Germany; jana.nano@helmholtz-muenchen.de; 7German Diabetes Center, 85764 München-Neuherberg, Germany; 8Institute of Cardiovascular Science, University College London, London WC1E 6BT, UK; 9MRC Centre for Environment and Health, Department of Epidemiology and Biostatistics, School of Public Health, Imperial College London, London SW7 2BX, UK; 10Centre for Inflammation Research and Translational Medicine (CIRTM), College of Health and Life Sciences, Brunel University London, Uxbridge UB8 3PH, UK; 11Department of Medicine, Internal Medicine, Lausanne University Hospital (CHUV), University of Lausanne, 1011 Lausanne, Switzerland; Pedro-Manuel.Marques-Vidal@chuv.ch

**Keywords:** blood pressure, systolic blood pressure, hypertension, menopause, age at menopause, mendelian randomization analysis

## Abstract

Observational studies suggest that early onset of menopause is associated with increased risk of hypertension. Whether this association is causal or due to residual confounding and/or reverse causation remains undetermined. We aimed to evaluate the observational and causal association between age at natural menopause (ANM) and blood pressure traits in Caucasian women. A cross-sectional and one-sample Mendelian randomization (MR) study was conducted in 4451 postmenopausal women from the CoLaus and Rotterdam studies. Regression models were built with observational data to study the associations of ANM with systolic and diastolic blood pressure (SBP/DBP) and hypertension. One-sample MR analysis was performed by calculating a genetic risk score of 54 ANM-related variants, previously identified in a genome-wide association study (GWAS) on ANM. In the two-sample MR analysis we used the estimates from the ANM-GWAS and association estimates from 168,575 women of the UK Biobank to evaluate ANM-related variants and their causal association with SBP and DBP. Pooled analysis from both cohorts showed that a one-year delay in menopause onset was associated with 2% (95% CI 0; 4) increased odds of having hypertension, and that early menopause was associated with lower DBP (β = −1.31, 95% CI −2.43; −0.18). While one-sample MR did not show a causal association between ANM and blood pressure traits, the two-sample MR showed a positive causal association of ANM with SBP; the last was driven by genes related to DNA damage repair. The present study does not support the hypothesis that early onset of menopause is associated with higher blood pressure. Our results suggest different ANM-related genetic pathways could differently impact blood pressure.

## 1. Introduction

Hypertension is the leading modifiable risk factor for cardiovascular diseases and mortality [1]. By 2015, the estimated rate of annual deaths associated with SBP of at least 140 mm Hg or higher was 106.3 per 100,000 persons [2]. Sex partially explains the differences in BP and hypertension across populations [3]. In general, men have a higher prevalence of hypertension compared with age-matched premenopausal women; however, after menopause women reach a similar prevalence [4].

Menopause onset is considered an independent marker of cardiovascular disease and mortality risk in women [5]. Early onset of menopause (e.g., <45 years), has been associated with high BP in several populations [6,7] and could increase cardiovascular disease risk observed in this group of women [5,6]. Nevertheless, the results of the association between age at menopause and BP traits are not consistent across studies [6,8,9], and causal association of age at menopause with BP has not been clarified [6]. Further, the association between early menopause and cardiovascular disease is independent of BP, suggesting other pathways (e.g., reverse causation) could explain the association [5,10].

The use of different approaches to address the causal association between ANM and BP traits is important. Mendelian randomization (MR) uses genetic variants as instrumental variables to provide evidence of causal relations in observational data. Currently, 54 single nucleotide polymorphisms (SNPs) have been associated with ANM in a large GWAS of women of European ancestry [11].

In this study, we aimed to evaluate the observational association between ANM and BP traits in Caucasian women. Next, we used the genetic loci associated with ANM to calculate an ANM genetic risk score, and through one- and two-sample MR analyses we evaluated whether there is a causal association between ANM and BP traits.

## 2. Methods

### 2.1. Study Population

We used data from the CoLaus study [12] and Rotterdam Study [13] for the observational analysis and one-sample MR; the design of these studies has been described elsewhere [12,13]. For the two-sample MR analysis we used data from the UK Biobank, a prospective study that recruited 500,000 participants, aged 40–69 years, from across the United Kingdom [14]. More details are provided in the Appendix A.

### 2.2. Population for Analyses

Only women reporting natural menopause were included in the observational and one-sample MR. Of 2875 women eligible in CoLaus, and of 2874, 1693, and 2249 eligible women in RS-I-3, RS-II-1, and RS-III-1, respectively, 4451 women were included for this analysis. Reasons of exclusion are depicted in the Appendix A and include being in perimenopause, experiencing non-natural menopause, and unavailable data on ANM, BP traits, or genetic data. To perform the two-sample MR we additionally used the information from the association analyses with SBP and DBP from the UK Biobank, which was restricted to 168,575 unrelated pre and postmenopausal women with European ancestry, whose variants had an information score ≥ 0.8 and minor allele frequency (MAF) ≥ 1%.

### 2.3. Assessment of Variables

In both CoLaus and RS, during the interview, women reporting being in postmenopause reported whether menopause was natural or due to other causes (e.g., hysterectomy), and the age at menopause. For women with natural menopause, ANM was defined as self-reported age at the time of last menstruation. In RS, menopause women were defined as women who reported the absence of menstrual periods for 12 months [15], while in CoLaus menopause was defined as the cessation of menses.

In CoLaus, BP was measured thrice on the left arm with an appropriately sized cuff, after at least a 10 min rest in the seated position using an Omron^®^ HEM-907 automated oscillometric sphygmomanometer. In RS, BP was measured twice on the right arm (cuff size of 32 × 17), after a resting period of 5 min in the seated position using a random-zero sphygmomanometer. In the UK Biobank, BP was measured twice in a seated position after 2 min of rest using an Omron HEM-7015IT digital BP monitor and an appropriate cuff [16]. SBP was recorded at the appearance of sounds (first-phase Korotkoff) and DBP at the disappearance of sounds (fifth-phase Korotkoff).

In all cohorts, SBP and DBP were calculated as the average of two measurements, in CoLaus only the last two measurement were used. In CoLaus and RS, hypertension was defined as a SBP ≥ 140 mm Hg, and/or DBP ≥ 90 mm Hg, and/or the use of antihypertensive medication. Medication use information was based on the home interview in RS and self-filled questionnaires in CoLaus. Antihypertensive medication use was defined as diuretics, β-blockers, angiotensin-converting enzyme inhibitors, and calcium channel blockers. In the RS, a physician also ascertained the indication for which the medication had been prescribed. Assessment of covariates is presented in the Appendix A [16].

### 2.4. Genotyping

In CoLaus, nuclear DNA was extracted from whole blood for whole genome scan analysis and genotyping was performed using the Affimetrix 500-K SNP chip, as recommended by the manufacturer [17,18,19]. In the RS, genotyping was conducted, in self-reported white participants, using the Illumina 550 K array, genotyping details are provided in the Appendix A. In the UK Biobank, 488,377 participants were genotyped and had extensive phenotypical information. DNA was extracted from stored blood samples collected from participants during their visit to an assessment center at the UK biobank. The genotype was performed by the Affymetrix Research Services Laboratory on 106 sequential batches [14].

### 2.5. SNPs Selection and the Genetic Risk Score

SNPs were selected based on the most extensive and recent report of association with self-reported ANM from the GWAS that included 33 studies and 69,360 women of European ancestry, where 54 SNPs were associated with ANM at the genome-wide significance level (*p* < 5 × 10^−8^), with MAF ranging from 7 to 49% and explaining 6% of the variance in ANM [11]. Based on this GWAS’s estimates and the genetic information of CoLaus and RS participants, we calculated a weighted genetic risk score (GRS) [20,21] for each study participant assuming each SNP to be independently associated with ANM according to an additive genetic model (Appendix A), GRS details are provided in the Appendix A.

### 2.6. Statistical Analyses

#### 2.6.1. Cross-Sectional Analyses

Multivariable linear and logistic regression models were used to assess the association of ANM (continuous) with SBP, DBP, and the presence of hypertension in CoLaus, RS-I-3, RS-II-1, and RS-III-1. For each study, we also compared levels of BP and prevalence of hypertension by categories of ANM (early, 40–44 years; intermediate, 45–49 years; normal, 50–54 years (reference); and late ≥55 years). Linear trends across ANM categories were tested by adding ANM categories as a continuous variable in the models. We constructed four models: model 1 included antihypertensive medications for SBP and DBP as outcomes; model 2 incorporated the variables in model 1 and chronological age; model 3 additionally included smoking, alcohol consumption, and educational level; and model 4 additionally included eGFR, BMI, history of cardiovascular diseases, total cholesterol, prevalent diabetes, statins, and hormone therapy. Interactions, model evaluations, log transformation of BP traits (Appendix A), and non-linear (spline) models were performed [22], as explained in detail in the Appendix A. To adjust for potential biases associated with missing data from the covariates, we used the multiple imputation procedure (*n* = 10 imputations) in the RS’s datasets (Appendix A).

To evaluate the individual effect of previous reported confounders of the association between ANM and BP, we conducted several sensitivity analyses by excluding women (a) reporting use of hormone therapy; (b) aged ≥ 65 years; (c) reporting a history of cardiovascular disease; (d) reporting menopause > 10 years ago; (e) with hypertension; (f) with hypertension defined as use of antihypertensive medication only; and (g) eGFR < 60 mL/min/1.73 m^2^. We also further adjusted by time since menopause and compared the estimates obtained in the imputed vs. the non-imputed database in the RS. Results across studies were summarized using the fixed effects model. In addition, the results from the random effects model were provided as a sensitivity analysis. A *p*-value < 0.05 was considered as statistically significant.

#### 2.6.2. One-Sample Mendelian Randomization

We performed linear regression and logistic regression analyses to examine the association between the GRS-ANM (either as a continuous variable or in quintiles) and BP traits. Mendelian randomization assumptions were explored (see Appendix A) [19,20]. Additionally, to further explore the role of the DNA damage response (DDR) pathway, previously identified in the reference GWAS [11], we constructed two sub-GRS groups according to the SNPs belonging to the DDR pathway and SNPs related to other pathways and repeated the analyses.

The 2-stage least squares (2SLS) regression [23] was applied using Stata V.15.1 (Stata Corp, College Station, TX, USA) command ivregress and control function estimation for the binary trait (hypertension) [24]. The 2SLS estimation proceeds by first fitting the regression of exposure (ANM) on the instrument (GRS-ANM) and then assessing the association of ANM with each outcome (SBP, DBP, and hypertension) on the fitted values from the first-stage regression. In order to generate estimates comparable with those from the observational regressions, we included age, eGFR, antihypertensive medication, BMI, drinking status, diabetes, history of cardiovascular diseases, total cholesterol, smoking status, statin use, hormone therapy, and education level as covariates.

#### 2.6.3. Two-Sample Mendelian Randomization

Additionally, we performed a two-sample MR to evaluate the casual association of ANM on SBP and DBP using the summary statistics of the GWAS of ANM [11] and UK Biobank estimates. ANM-SNPs statistics were extracted from Day et al., GWAS [11]. SNP rs4886238 was excluded from our final analysis because it did not fulfill the quality criteria. Using data from the UK Biobank, outcome summary statistics for 53 ANM-related SNPs and BP traits were generated by carrying out association analyses with SBP and DBP, adjusted for the first four principal components for the genetic variability of the genome, age at baseline, and the genotyping array used. The UK Biobank study was not included in the GWAS of ANM [11], and therefore fulfilled the independency sample criteria of two-sample MR.

We applied four MR methods including an inverse-variance weighted average approach (IVW), MR-Egger regression, weighted median approach, and MR-PRESSO [25]. Models are explained in the Appendix A. Linkage disequilibrium (LD) of the SNPs was assessed against a reference European population from where an LD matrix of the evaluated SNPs was created; LD values are with respect to the major alleles. SNPs with values greater than 0.001 LD R-squared and with the highest *p* value were removed. Harmonization of the reported alleles was performed according to previous guidelines [26] (Appendix A). To consider the association as causal, three of the four implemented methods should provide coherent results [27].

The heterogeneity of the estimates was evaluated through Cochran’s Q test and scatter plot. Second, directional pleiotropy was examined using funnel plots. Third, a leave-one-out sensitivity analysis was performed to evaluate the effect of each of the variants on the causal estimate. Fourth, the analysis was repeated by splitting the SNPs into two groups: one based only on SNPs related to DDR genes (*n* = 37) and the other based on genes other than DDR [11]. Fifth, to address potential pleiotropy, the SNPs previously reported as associated with BMI and menarche [28] were excluded. Finally, an analysis not adjusting for correlation and including the weak SNPs was undertaken. Analyses were conducted using Stata V.15.1 (Stata Corp, College Station, TX, USA) and R software (version 3.6.3; R Foundation for Statistical Computing, Vienna, Austria) (Mendelian Randomization [29] and Two-Sample MR [30] packages).

## 3. Results

The baseline characteristics of the 4451 participants who satisfied the inclusion criteria from both studies are shown in Appendix A. The reported early onset of menopause (<45 years) varied from 6.8 in RS-III-1 to 9.7% in CoLaus. Prevalence of hypertension ranged between 43.2 in CoLaus and 69.3% in RS-I-3, and SBP ranged between 131 ± 18.7 mm Hg in CoLaus and 144 ± 21.3 mm Hg in RS-I-3.

No association was found between ANM as a continuous variable and SBP in CoLaus, RS-I-3, and RS-II-1, while in RS-III-1, the one-year increase in ANM was associated with 0.45 mm Hg increase in SBP (95% CI: 0.11; 0.78) (Figure 1, Appendix A). The pooled analysis showed increased SBP with increasing menopausal age (β = 0.11, 95% CI −0.04; 0.25), albeit the association was not significant. Early menopause, compared to menopause between 50–54 years, was associated with lower DBP (β = −1.31 mm Hg, 95% CI: −2.43; −0.18), while no association was found between other ANM categories and DBP (Figure 1). The pooled analysis of ANM as a continuous variable showed later onset of menopause to be associated with higher odds of developing hypertension (odds ratio (OR): 1.02, 95% CI: 1.00; 1.04) while no association was found for the ANM categories and hypertension (Figure 1).

Interaction of ANM and BMI was not significant in any of the analyses (Appendix A). The main findings did not change when time since menopause was included in the models or when the analyses were restricted to the non-imputed dataset of the RS cohorts (Appendix A). We did not find evidence of a non-linear association of ANM and BP traits in the spline’s evaluation (Appendix A). After the exclusion of (a) 827 women reporting use of hormone therapy, early onset of menopause was associated with lower SBP (β = −2.95, 95% CI −5.28; −0.62) and DBP (β = −1.29, 95% CI −2.56; −0.02) and decreased odds of hypertension (OR = 0.71, 95% CI: 0.53; 0.94) (Appendix A). Similarly, after the exclusion, of (b) 2052 women aged > 65 years; (c) 350 women reporting a history of cardiovascular diseases; and (e) 2548 women who had hypertension (for SBP and DBP as outcome), early onset of menopause was associated with lower DBP, while no differences were observed after exclusion of the other groups of women mentioned in the Methods Section (Appendix A).

### 3.1. One-Sample MR

The GRS-ANM was associated with observed ANM and explained between 1.4% and 3.4% of the ANM variance in the included cohorts; F statistic values were between 11.15 and 40.63 (Appendix A). We found no association between GRS-ANM and SBP, DBP, or hypertension in either of the cohorts (Appendix A).

In the MR analysis using the GRS-ANM as the instrumental variable and pooling the estimates through meta-analysis, we found no evidence of a causal association between ANM and BP traits in the crude and adjusted analyses (Table 1). Neither individual genetic variants nor the GRS were associated with potential confounders (Appendix A). Similarly, we found no evidence of association between the 54 SNPs ANM and BP traits (Appendix A). In the “leave-one-out” analyses, we observed that in RS-I-3, RS-II-1, and RS-III-1, the GRS-ANM excluding SNP rs1054875 showed a weaker association with the observed ANM, while excluding the others SNPs from the GRS-ANM did not materially change the association. However, the genetic risk score without including SNP rs1054875 was evaluated with the instrumental variable approach, and the findings were consistent with the main analysis (Appendix A). Finally, we did not observe changes in the pooled casual estimates when we evaluated a genetic risk score with 37 SNPs related to DDR genes. Nevertheless, the pooled results of the GRS with the remaining 17 SNPs non-related to DDR genes showed a marginal causal association of later ANM on SBP (β = −2.16, 95% CI: −4.30; −0.02) (Appendix A).

### 3.2. Two-Sample MR

In the UK Biobank, SNP rs4886238 did not reach quality control; 13 SNPs were excluded due to LD with other variants, leaving 40 independent SNPs for the analysis. The results of the two-sample MR were consistent with the findings of non-causal association between ANM and DBP observed in the one-sample MR analyses. We found evidence of causal effect of ANM on SBP (*p*-value < 0.05) in three of the four methods implemented (weighted median β = 0.36; 95% CI: 0.18; 0.53, IVW β = 0.25; 95% CI: 0.04; 0.45 and MR-PRESSO β = 0.23; 95% CI: 0.20; 0.26), while MR-Egger did not support a causal effect (β = 0.28; 95% CI: −0.20; 0.75) and did not indicate potential directional pleiotropy in either DBP or SBP (*p*-value for intercept 0.669 and 0.884, respectively). Significant heterogeneity for ANM was found based on the Cochran’s test < 0.001 (Table 2), and asymmetrical distribution was observed in the funnel plots (Appendix A).

The scatter plots suggested that the SNP rs16991615 had a dominant effect on the analysis (Appendix A). However, in the leave-one-out analyses, we did not find any genetic variant driving the overall effect of the ANM on SBP/DBP (Appendix A). The exclusion of the four outlier SNPs (Appendix A) identified in the MR-PRESSO did not change the findings (Table 2). The analyses of SNPs related to the DDR genes was coherent with the main analysis; in contrast, the analysis of the non-DDR genes showed no causal association with BP traits (Appendix A). Similarly, estimates did not change after the exclusion of possible pleiotropic SNPs, (six related to adult BMI and two with age at menarche) (Appendix A). No evidence of casual effects of ANM on SBP/DBP was observed after the inclusion in the analysis of the full set of the 53 SNPs (Appendix A).

## 4. Discussion

Our results do not support the hypothesis that early onset of menopause is associated with higher BP, but they suggest a potential positive causal association between ANM and SBP, which could depend on specific genetic pathways related to ANM. No causal association was found for the association of ANM and DBP.

In the observational analysis, after adjustment for a broad range of confounding factors, we found that early onset of menopause was not associated with an adverse blood pressure profile. In contrast, there was an indication that later onset of natural menopause could be associated with higher blood pressure levels and higher odds of hypertension, which was further supported by the two-sample MR analysis indicating a potential causal association. Additionally, ANM was associated with a small but statistically significant increase in the risk of hypertension. In general, age at menopause has been associated with hypertension in different populations, but results have not been consistent [6,31,32,33,34]. A large study of 7893 women reported that age at menopause was associated with high odds of having hypertension, but this association was not significant after further adjustment for BMI [35]. Similar findings were reported in a meta-analysis of observational studies evaluating the association between early menopause and hypertension matched by age or BMI, with a pooled estimate being marginally significant (OR = 1.13; 95% CI: 1.00; −1.29) [6]. In addition to confounding, reverse causation can also play a role in explaining the contradictory findings. Conditions such as hypertension, coronary heart diseases, and stroke can be related to menopause onset [10]. A physiological explanation for the effect of hypertension is the possible decline in ovarian blood flow, which could lead to follicle loss and substantial decrease in ovarian reserve with the acceleration of the onset of menopause [36,37,38].

However, in the current study, the two-sample MR analysis, which had higher power than the one-sample MR to detect small effects between ANM and SBP, suggested a causal positive effect of age at menopause on SBP. Nevertheless, due to the complexity of pathways associated with the onset of menopause and the effects on the hormonal and vascular systems, it is currently unclear how later onset of menopause could influence adverse changes in BP levels. The potential causal association of DDR genes with SBP observed in our study could present a possible pathway linking ANM with BP. Molecular damage such as DNA fragmentation is frequent in hypertensive patients [39]. On the other hand, there is evidence to show that prolonged exposure to estrogen [40], which could be the case in women experiencing late menopause, could lead to DNA damage. DNA damage induced by estrogen is considered an important risk factor for breast cancer [40,41]. Increased BP and later onset of menopause are both associated with increased risk of breast cancer [42]. In addition, DNA damage could be induced by stress hormones such as glucocorticoid and catecholamines, which are affected by menopause [40,43]. To date, DDR variants have not been associated with BP traits, and the role of DNA repair in BP is not fully understood. Future studies are needed to elucidate the potential role of DNA repair as a potential underlying mechanism behind the association of ANM and BP and the potential interaction of sex hormones with DNA repair and BP in women.

### 4.1. Strengths and Limitations

Major strengths of our study include the use of instrumental genetic variables strongly associated with ANM that have been replicated in other studies, the combination of data from two large comprehensive studies, and accounting for multiple confounding factors. Another strength is the use of aggregate data from the largest GWAS of ANM and UK Biobank, which takes advantage of the large sample size and similarity between the studied populations.

Nevertheless, some important limitations are worth considering. ANM related SNPs explain only a small proportion of the estimated heritability of ANM, and the study sample was restricted to Caucasian women, which limits the extrapolation of the results to other racial/ethnic populations. Additionally, age at menopause was assessed retrospectively several years after the menopause occurred, which could lead to a recall bias [44]. However, it is assumed that this error was not differential. Furthermore, due to the cross-sectional design, the reported estimates could be influenced by residual confounding factors related to hormone therapy, chronological age, and time since menopause; similarly, differences in sample size and hypertension prevalence could also be related to the heterogeneity of the estimates in our analysis. One-sample MR analysis might have limited power to detect small effects; moreover, our analysis inherits all the limitations that the MR itself has as a method (e.g., buffering mechanisms) [45,46,47,48]. In the two-sample MR we could not explore further sensitivity analyses (i.e., evaluate the genetics associations with other reproductive traits associated with BP traits that are out of the causal pathway of the ANM). Altogether, these limitations could lead to the existence of pleiotropic effects. However, the MR-Egger intercept was indicative of validity of the genetic variants analyzed. Additionally, we performed a “leave-one-out” analysis, which did not change the findings.

The MR-Egger causal effect was not significant; however, the intercept was not indicative of directional pleiotropy. Therefore, even the causal estimate in the MR-Egger method was imprecise; it was coherent with the estimates from the other approaches implemented, which does not contradict the causal relation between ANM and SBP [49]. Additionally, the MR-Egger analysis is sensitive to outliers [49]; in our study we identified four outlying SNPs, whose effects were corrected in the MR-PRESSO analysis, and the causal effect remained significant.

### 4.2. Future Research and Implications

The results presented here support the importance of considering age at menopause as an exposure in preclinical and clinical studies [50]. Findings suggest that the mechanisms behind the observed associations are more complex and need better evidence supporting the identification of predictors or risk factors to bring timely interventions to women [51]. Therefore, future studies should focus on the evaluation of the DNA damage response pathways, ageing process, and their role in the onset of menopause and progression of cardiovascular disease in women. Another potential pathway linking ANM with BP could be epigenetic modifications such as DNA methylation of cytosine residues in CpG dinucleotides and histone modification. Epigenetic mechanisms are associated with both ANM and BP traits [52], and future studies should explore epigenetic modifications related to menopause onset and whether the identified epigenetic signatures can explain the association between ANM and BP. In addition, large observational cohort studies shall explore whether use of antihypertensive medication and changes in SBP and DBP prior to menopause are prospectively associated with onset of menopause, and complementary MR analysis could shed light on whether the effect is casual, given there is an association. Finally, it is necessary to improve the assessment of age at menopause by assessing it prospectively and identifying biomarkers associated with the onset of menopause to achieve greater accuracy.

In conclusion, findings of our study provide additional support that early onset of menopause is not associated with a higher BP profile. In contrast, later onset of menopause might be causally associated with adverse systolic blood pressure levels, and the DNA damage repair pathways might be potential mechanisms underlying the association.

## Figures and Tables

**Figure 1 jcm-10-04299-f001:**
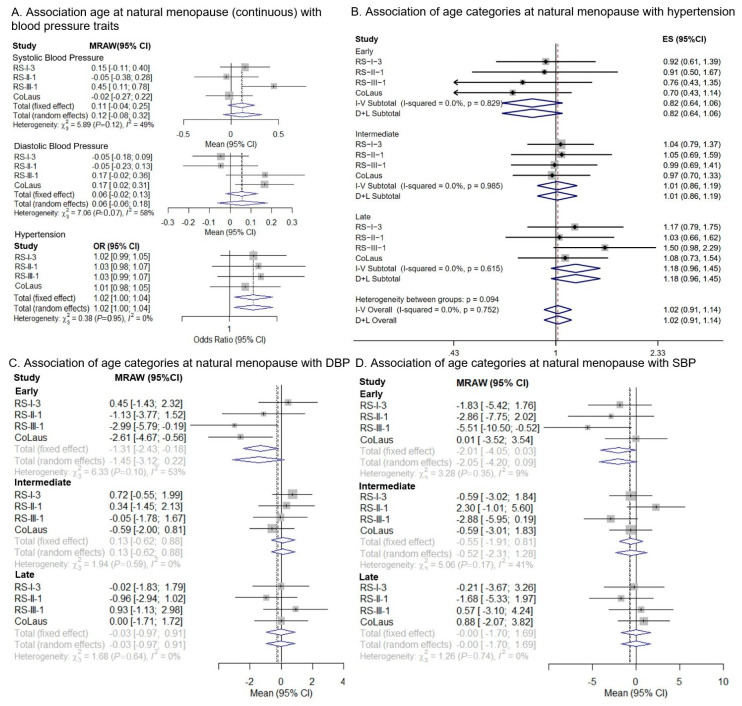
(**A**) Forest plot of observational estimates for the association between age at natural menopause (continuous) and blood pressure traits; (**B**) Forest plot of observational estimates for the association between categories of age at natural menopause and hypertension; (**C**) Forest plot of observational estimates for the association between categories of age at natural menopause and diastolic blood pressure (DBP); (**D**) Forest plot of observational estimates for the association between categories of age at natural menopause and systolic blood pressure (SBP). A fixed-effects meta-analysis model was used in all cases. Models adjusted for age and glomerular filtration rate, body mass index, total cholesterol, drinking status, education level, smoking status, diabetes, history of cardiovascular diseases, statin use, hormone therapy, and use of antihypertensive medication. Hypertension was defined as a systolic BP ≥ 140 mm Hg and/or diastolic BP ≥ 90 mm Hg, and/or the use of antihypertensive medication. Early 40–44, intermediate 45–49, normal 50–54, late ≥55.

**Table 1 jcm-10-04299-t001:** Causal estimates derived from instrumental variable analysis for a year increase in ANM and risk of blood pressure traits. Individual participant data were used for the analyses.

	CoLaus (*n* = 1139)		RS-I-3 (*n* = 1603)		RS-II-1 (*n* = 790)		RS-III-1 (*n* = 919)		Meta-Analysis	
Variable	β (95% CI)	*p*	β (95% CI)	*p*	β (95% CI)	*p*	β (95% CI)	*p*	β (95% CI)	I^2^
Crude										
Systolic BP	−0.80 (−4.17; 2.61)	0.65	−1.89 (−4.23; 0.45)	0.11	0.68 (−2.50; 3.87)	0.68	0.93 (−1.50; 3.38)	0.45	−0.36 (−1.72; 1.01)	5.9%
Diastolic BP	−0.02 (−1.85; 1.81)	0.98	−0.57 (−1.72; 0.58)	0.33	0.48 (−1.06; 2.02)	0.54	0.41 (−0.95; 1.77)	0.56	−0.01 (−0.71; 0.70)	0.0%
* Hypertension	1.09 (0.76; 1.54)	0.68	0.90 (0.71; 1.13)	0.35	1.01 (0.81; 1.47)	0.55	0.97 (0.76; 1.24)	0.84	0.97 (0.85; 1.11)	0.0%
Adjusted	(*n* = 1137)									
Systolic BP	−0.84 (−3.72; 2.05) ^†^	0.570	−1.86 (−4.13; 0.41)	0.11	−0.76 (−3.92; 2.38)	0.63	0.40 (−2.06; 2.87)	0.75	−0.82 (−2.13; 0.50)	0.0%
Diastolic BP	0.25 (−1.40; 1.90)	0.77	−0.56 (−1.68; 0.56)	0.33	0.27 (−1.42; 1.98)	0.75	0.10 (−1.29; 1.48)	0.89	−0.10 (−0.81; 0.60)	0.0%
* Hypertension	1.03 (0.70; 1.50)	0.89	0.94 (0.73; 1.20)	0.63	1.02 (0.73; 1.43)	0.90	0.92 (0.71; 1.21)	0.56	0.96 (0.83; 1.11)	0.0%

*n* = number of participants; Adjusted = Fully adjusted model: adjusted for antihypertensive medication; body mass index, drinking status, diabetes, history of cardiovascular diseases, total cholesterol, smoking status, statin use, hormone therapy, education level. β estimates; CI = confidence interval; *p* = *p*-value * Odds ratio ^†^ 1137 observations.

**Table 2 jcm-10-04299-t002:** Causal estimates of age at natural menopause with blood pressure traits (systolic and diastolic blood pressure) using methods implemented in the MR-Base with data from the GWAS of ANM and UK Biobank.

		40 SNPs				
Outcome	Method	β	(95% CI)	*p*-Value	*p* _h_	Q-Statistics
Diastolic BP	Weighted median	0.03	−0.08; 0.14	0.554	<0.001	141.48
	Inverse variance Weighted	0.05	−0.08; 0.17	0.460		
	MR-PRESSO	0.05	0.03; 0.06	0.354		
	MR-Egger	0.10	−0.19; 0.40	0.481		
	MR-Egger, intercept	−0.01	−0.07; 0.05	0.669		
Systolic BP	Weighted median	0.36	0.18; 0.53	0.000	<0.001	138.80
	Inverse variance weighted	0.25	0.04; 0.45	0.020		
	MR-PRESSO	0.23	0.20; 0.26	0.009		
	MR-Egger	0.28	−0.20; 0.75	0.253		
	MR-Egger, intercept	−0.01	−0.10; 0.09	0.884		

CI = Confidence interval; *p*_h_, *p*-value for heterogeneity, exact *p*-value Cochran’s Q test; Diastolic BP 1.526 × 10^−13^, exact *p*-Value Cochran’s Q test; Systolic BP 4.116 × 10^−13^.

## Data Availability

All the data relevant to this study are freely available in the paper and Appendix A. The study participants of CoLaus have not provided consent to publicly share the individual level data included in this study. Information related to data access is available to qualified, interested researchers at https://www.colaus-psycolaus.ch/professionals/how-to-collaborate/ (accessed on 20 September 21). All responses to data sharing requests must comply with the ethical and legal constraints of Switzerland. Individual level data from the Rotterdam Study participants can be made available by request and approval of the Rotterdam Study Management Team (secretariat.epi@erasmusmc.nl).

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
