# Peer review of "Age at Natural Menopause and Blood Pressure Traits: Mendelian Randomization Study"

_jcm, 2021, doi:10.3390/jcm10194299_

Round 1

Reviewer 1 Report

It is a very detailed subject treated, with adequate statistical analysis for this type of research
Major strengths of the study are instrumental genetic variables strongly associated with ANM and the use of aggregate data from the largest
GWAS of ANM and UK Biobank.
The limitation - "age at menopause was assessed retrospectively
several years after the menopause occurred" will be avoid in future research.

Author Response

Comments to the Author

It is a very detailed subject treated, with adequate statistical analysis for this type of research

Major strengths of the study are instrumental genetic variables strongly associated with ANM and the use of aggregate data from the largest GWAS of ANM and UK Biobank.

RESPONSE. We thank the reviewer for the comments. We hope that the changes we made reflect upon the reviewer’s comments and suggestions.

The limitation - "age at menopause was assessed retrospectively several years after the menopause occurred" will be avoid in future research.

DONE: We thank the reviewer for presenting this important limitation. We included a recommendation in the paragraph of future research and implications.

Page 17, Lines 394-395:“Finally, it is necessary to improve the assessment of age of menopause by assessing it prospectively and identifying biomarkers associated with the onset of menopause to achieve greater accuracy.”

Reviewer 2 Report

In this paper, contrary to initially expected, the authors showed a lack of relationship between early menopause and higher blood pressure in postmenopausal women. However, it shows a small positive association between genetic traits associated to menopause and hypertension in the same population. Neither any interaction between the age of menopause and body mass index was observed.
Although "negative", these results are very relevant because they establish the lack of association between traditionally connected factors as early menopause and higher blood pressure as a mechanism of cardiovascular damage.  As the authors explain in the text, the explanation may be an inverse causality since cardiovascular disease decreases ovarian blood flow, which is one of the causes of menopause. 

Nevertheless, the results underlined the connection between menopause and hypertension by genetic traits related to DNA damage by Mendelian randomization of genes related to menopause onset. 

Minor commentaries:
The study included two large populations with European ancestry, and results could not be generalized to other populations. This issue may be considered a limitation since hypertension is more related to Afroamerican subjects, a population in whom hypertension-related genetic traits are more frequent. 
How do the authors explain the heterogeneity of results in the association between age of menopause, systolic blood pressure and diastolic blood pressure? (Figure 1A)

Author Response

Comments to the Author

In this paper, contrary to initially expected, the authors showed a lack of relationship between early menopause and higher blood pressure in postmenopausal women. However, it shows a small positive association between genetic traits associated to menopause and hypertension in the same population. Neither any interaction between the age of menopause and body mass index was observed.

Although "negative", these results are very relevant because they establish the lack of association between traditionally connected factors as early menopause and higher blood pressure as a mechanism of cardiovascular damage.  As the authors explain in the text, the explanation may be an inverse causality since cardiovascular disease decreases ovarian blood flow, which is one of the causes of menopause.

Nevertheless, the results underlined the connection between menopause and hypertension by genetic traits related to DNA damage by Mendelian randomization of genes related to menopause onset.

RESPONSE. We thank the reviewer for the positive feedback on our manuscript. We have now addressed the issues highlighted by the reviewer.

 (1) The study included two large populations with European ancestry, and results could not be generalized to other populations. This issue may be considered a limitation since hypertension is more related to Afroamerican subjects, a population in whom hypertension-related genetic traits are more frequent.

 DONE. We thank the reviewer for this important comment. We have emphasized this limitation in the limitations section.

Page 17, Lines 360-361: “the study sample was restricted to Caucasian women, which limits the extrapolation of the results to other racial/ethnic populations

 (2) How do the authors explain the heterogeneity of results in the association between age of menopause, systolic blood pressure and diastolic blood pressure? (Figure 1A)

 DONE. We thank the reviewer for this important question. We included possible sources of heterogeneity in the limitations section.

Page 17, Lines 363-366 Furthermore, due to the cross-sectional design, the reported estimates could be influenced by residual confounding factors related to hormone therapy, chronological age, and time since menopause; similarly, differences in sample size and hypertension prevalence could be also related to the heterogeneity of the estimates in our analysis.
